REGISTERED REPORT PROTOCOL

# Analysis of biliary MICRObiota in hepatoBILIOpancreatic diseases compared to healthy people [MICROBILIO]: Study protocol

**Fernanda Sayuri do Nascimento[1], Milena Oliveira Suzuki[1], João Victor Taba[1], Vitoria Carneiro de Mattos[1], Leonardo Zumerkorn Pipek****[1], Eugênia Machado Carneiro D'Albuquerque[2], Leandro Iuamoto[3], Alberto Meyer****[4]\*, Wellington Andraus[4], João Renato Rebello Pinho[4], Eduardo Guimarães Hourneaux de Moura[4], João Carlos Setubal[5], Luiz Augusto Carneiro-D'Albuquerque[4]**

**1** Faculty of Medicine FMUSP, University of São Paulo, São Paulo, São Paulo, Brazil, **2** Santa Marcelina Faculty, São Paulo, São Paulo, Brazil, **3** Colaborator, Center of Acupuncture, Department of Orthopaedics and Traumatology, University of Sao Paulo School of Medicine, São Paulo, São Paulo, Brazil, **4** Department of Gastroenterology, Hospital das Clínicas, HCFMUSP, São Paulo, São Paulo, Brazil, **5** Department of Biochemistry, Institute of Chemistry, University of São Paulo, São Paulo, São Paulo, Brazil

\* alberto.meyer@usp.br

This is a Registered Report and may have an associated publication; please check the article page on the journal site for any related articles.

## Abstract

### Background

The performance of the microbiota is observed in several digestive tract diseases. Therefore, reaching the biliary microbiota may suggest ways for studies of biomarkers, diagnoses, tests and therapies in hepatobiliopancreatic diseases.

### Methods

Bile samples will be collected in endoscopic retrograde cholangiopancreatography patients (case group) and living liver transplantation donors (control group). We will characterize the microbiome based on two types of sequence data: the V3/V4 regions of the 16S ribosomal RNA (rRNA) gene and total shotgun DNA. For 16S sequencing data a standard 16S processing pipeline based on the Amplicon Sequence Variant concept and the qiime2 software package will be employed; for shotgun data, for each sample we will assemble the reads and obtain and analyze metagenome-assembled genomes.

### Results

The primary expected results of the study is to characterize the specific composition of the biliary microbiota in situations of disease and health. In addition, it seeks to demonstrate the existence of changes in the case of illness and also possible disease biomarkers, diagnosis, interventions and therapies in hepatobiliopancreatic diseases.

### Trial registration

NCT04391426. Registered 18 May 2020, https://clinicaltrials.gov/ct2/show/NCT04391426.

**Data Availability Statement:** All relevant data are within the paper and its Supporting Information files.

**Funding:** The authors received no specific funding for this work.

**Competing interests:** The authors have declared that no competing interests exist.

**Abbreviations:** DNA, Deoxyribonucleic Acid; rRNA, ribosomal Ribonucleic Acid; CRC, Colorectal Carcinoma; BPT, Biliopancreatic Tract; PDAC, Pancreatic Ductal Adenocarcinoma; ERCP, Endoscopic Retrograde Cholangiopancreatography; BMI, Body Mass Index; HIV, Human Immunodeficiency Virus; CT, Computed Tomography; MRI, Magnetic Resonance Imaging; LIM-03, Central Laboratory Division 3; QIIME 2, Quantitative Insights Into Microbial Ecology; ASV, Amplicon Sequence Variants; DADA2, Deficiency of Adenosine Deaminase 2; STAMP, Statistical Analysis of Taxonomic and Functional Profiles; GTDB-tk, Genome Taxonomic Database; PCA, Principal Component Analysis; FMUSP, Faculty of Medicine, University of São Paulo; CAAE, Certificate of Presentation and Ethical Appreciation.

## Introduction

Microbioma is the set of microorganisms that occurs naturally in a particular site, such as the human gastrointestinal tract. Typically, it has trillions of microbes, including fungi, viruses and bacteria [1], which coexist with human cells.

Normally, the microbiome bacteria interact with the epithelial barrier, with immune cells modulating their response, in addition to influencing local metabolism through their own metabolites. This maintains homeostasis [2]. Thus, an imbalance of the microbiota, such as the use of antibiotics or due to bacterial translocation, can lead to the development of diseases. It does this through the proliferation of pathogenic bacteria, for example, which can greatly affect the host and have potential pathological implications [2–4].

Studies have shown a close relation between dysbiosis and the outbreak of infections or chronic diseases. In 2019, Saus et al. [5] gathered data about the relation between the intestinal microbiota and the development of colorectal carcinoma (CRC), the most studied since the 1990s. Currently, it is known that in patients with this neoplasm there is a co-abundance of pro-inflammatory factors, opportunistic pathogens and other microbes. This is associated with metabolic dysfunction and the depletion of butyrate-producing bacteria, an important factor in intestinal homeostasis. With these studies, interest was raised in investigating other sites, such as the biliopancreatic tract (BPT). In 2015, Mitsuhashi et al. showed the association of the oral microbiota with the pancreatic carcinogenesis process. In periodontal diseases, *Fusobacterium* can be translocated via lympho-hematogenous pathways, leading to pancreatic dysbiosis. This would be associated with malignancy in the progression of pancreatic adenocarcinoma and worse prognosis [4].

Thus, research was conducted in rats, finding an association between the components of the tumour microbiota and the speed of progression of biliopancreatic disease [6]. In pancreases of rats and humans with pancreatic duct adenocarcinoma (PDAC), a greater abundance of *Malassezia spp.* was found compared to bowel or pancreas controls without the disease. Due to the presence of the fungus, there is greater activation of mannose-binding lectin (MBL) and, consequently, the complement cascade is activated, leading to greater inflammation in the pancreas, which accelerates tumour progression [6, 7].

Traditionally, BPT neoplasia are diagnosed at an advanced stage, despite the improvement in the quality of diagnostic imaging. For early diagnosis, Mendez et al. conducted an experiment with PDAC-mutated mice before they developed the disease. By DNA sequencing of the fecal microbiota bacteria, it was found that, with the progression of pancreatic carcinogenesis, there was a change in the bacterial composition. The metabolites of these bacteria associated with the tumour promote greater production of polyamines, which increases as the neoplasia develops. Then, the dosage of polyamines could be used as a biomarker to track the progression of adenocarcinoma [8]. The analysis of the fecal microbiota, therefore, would be a possibility for early diagnosis of PDAC, prompting research in human patients at high risk for carcinogenesis.

Among the therapies, the most commonly used treatments for neoplasia are chemotherapy and radiotherapy. In BPT carcinomas, however, these methods have low sensitivity [9], which can be attributed, according to studies, to tumor dysbiosis. As previously mentioned, unchanged symbiotic microbiota mediates the immune response. Thus its imbalance decreases the expression of genes related to inflammation, phagocytosis, antigen presentation and adaptive immune response. On the other hand, genes related to tissue development, cancer and metabolism are stimulated [10]. Thus, the use of chemotherapeutic drugs loses effectiveness due to this negative regulation of the immune system's anti-tumor capacity. An example of this is oxaliplatin, whose effect is to stimulate the production of reactive oxygen species to

promote DNA damage and tumor cell apoptosis. In mice injected with colon carcinoma cells, their cytotoxic effect decreased after being treated with antibiotics. [9, 10].

There have been several advances in the introduction of new chemical compounds that interfere in specific signalling pathways of carcinogenesis, also affected by the microbiome. Because of this association, microbial agents and their metabolites are being tested to develop treatments that can reduce the tumor and are potentially preventive [11]. This was observed in studies with species of *Lactobacillus*, which modulates the expression of some enzymes such as beta-glucuronidase. The action of the enzyme is reduced by bacilli. This acts in the disjunction of carcinogenic agents, converting pro-carcinogens into their active form [12]. Lenoir et al. also demonstrated that *L.casei* has anti-tumor properties by decreasing the T-reg response and increasing Th17, promoting a decrease in CRC in rats. Thus, the microbe proved to be a protector and a new therapeutic alternative to carcinoma [11, 13].

Even with current treatments, mortality in some groups of BPT malignancies remains high, with low survival rates. However, in recent studies in patients with pancreatic duct adenocarcinoma, it was found that the greater variety of the tumor microbiota and the predominance of specific bacterial genera are related to a longer survival time when treated surgically [6]. Considering this recent progress, improvement is expected for the coming decades.

Thus, the performance of the microbiota is observed in all clinical and pathological stages of carcinogenesis, from its development, diagnosis and treatment, including prognosis and survival. However, there is a lack of studies on biliary microbiota and its relation with hepatobiliopancreatic diseases. Therefore, further investigation is necessary, since researching the biliary microbiota may suggest ways for studies of biomarkers, diagnoses, interventions and therapies in hepatobiliopancreatic diseases.

## Aim

In this study, our aim will be to characterize the specific composition of the biliary microbiota in patients with hepatobiliopancreatic diseases compared to healthy controls, using 16S ribosomal RNA (rRNA) pyrosequencing methods.

## Materials and methods

### Medical costs

Medical costs or other cash payments to donors and families will no be offered and participation will be voluntary. All procedures will be performed in the public health system, at Hospital das Clínicas University of Sao Paulo School of Medicine (HCFMUSP), entirely free of charge.

### Study design and patients

This is a case-control study that will be carried out at the Department of Gastroenterology of Clinicas Hospital in the Faculty of Medicine of the University of São Paulo.

Patients who will undergo endoscopic retrograde cholangiopancreatography (ERCP) and donors previously selected to interventional liver transplantation will be recruited for the collection of bile, configuring case and control groups, respectively. The project outline is illustrated in Fig 1.

### External validation

All patients who will not be included, with a condition for which a procedure is planned, such as ERCP or hepatectomy for liver transplantation, will have their records noted and stored,

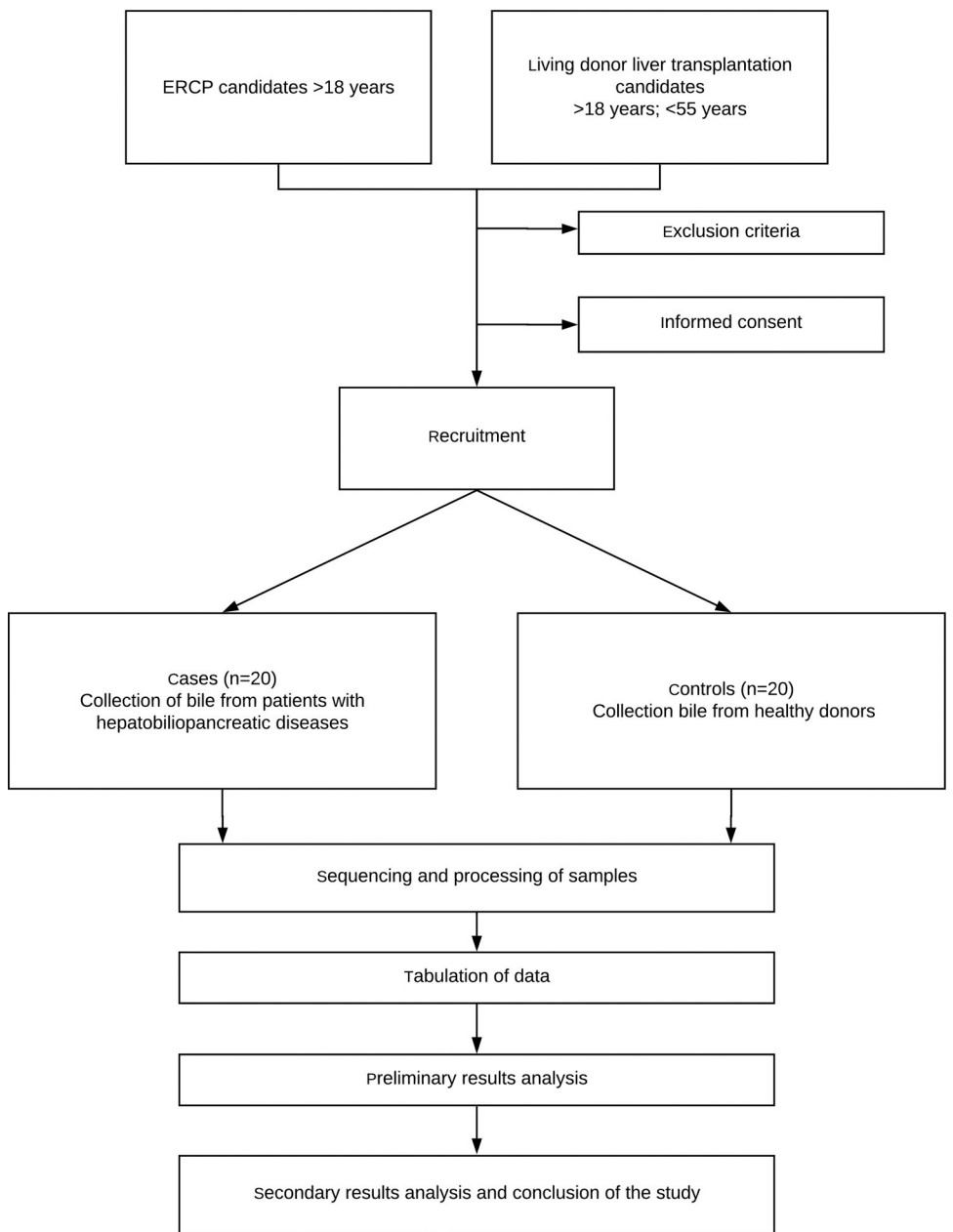

**Fig 1. Study scheme.**

including information on date, sex, age, Body Mass Index (BMI) and the reason for non-inclusion or exclusion.

## Patients who will undergo ERCP (case group)

### Inclusion criteria

- Patients over 18 years old

- Patients previously scheduled for ERCP

- Cannulation of the bile duct, via the transpapillary route, with the aid of a papillotome with an end kept sterile until contact with the papilla

## Exclusion criteria

- Use of antibiotics during ERCP or in the last 2 months prior to the procedure

- Emergency ERCP

- Pregnancy

- Uncorrected coagulopathy

History of previous ERCP will not be considered an exclusion criteria.

**ERCP technical description.** The procedure will be performed under conscious sedation or general anesthesia, at the discretion of the medical team responsible for carrying out the procedure.

Before the beginning of ERCP, the end of the papillotome will be covered with sterile surgical plastic (plastic cover for videolaparoscopy), in order to avoid its contamination during passage through the working channel of the duodenoscope and contact with the digestive tract, up to the greater duodenal papilla.

The duodenoscope will be introduced according to the usual technique and positioned in front of the greater duodenal papilla. The papillotome with a sterile end will then be passed through the working channel of the duodenoscope. Before attempting cannulation of the papilla, air will be injected through the papillotome injection channel to remove the sterile plastic from its end.

When performing cannulation of the bile duct, with the aid of endoscopic and radioscopic vision, the route of the papillotome injection until bile return will be aspirated with a sterile 5 ml syringe, to confirm the correct positioning of the instruments in the bile duct. Then, 1 ml of bile will be aspirated, which will be sent for analysis of the microbiota.

If there is contamination of the papillotome prior to contact with the papilla, the patient will be excluded from the protocol.

## Living liver transplantation donors (control group)

The correct selection of donors for living donor liver transplantation is essential not only to decrease the risk of complications for donors, but also to increase graft and recipient survival.

First, there must be ABO blood type compatibility. Then, the compatibility between the weight and height of the patient and the donor is analysed. Subsequently, the size of the liver to be donated is evaluated, and it is necessary to calculate the relation between the weight of the liver (donor) with the recipient. Finally, the entire anatomy of the donor and recipient is evaluated, such as veins, arteries and bile ducts.

## Inclusion criteria

- Patients over 18 years old up to 55 years old

- Previously selected patients with scheduled surgery

- BMI: 18 $kg/m^2$ to 28 $kg/m^2$

- Blood typing identical to the recipient

- Absence of significant medical, psychiatric problems or previous abdominal surgery

- Normal laboratory tests: liver function tests, blood count, coagulogram, pregnancy test and serology for hepatitis B, C and HIV

- Normal imaging exams: CT of the abdomen and pelvis with liver volume (remaining volume—30–40% of the total liver volume), MRI with cholangioresonance

## Exclusion criteria

- Use of antibiotics in the last 2 months prior to the procedure

- Pregnancy

- Uncorrected coagulopathy

è Description of the interventional liver transplantation technique

The hepatectomy procedure for related living donor liver transplantation, left lobe or right lobe will be performed under general anesthesia.

Started by a Makuuchi incision ("J" incision), followed by positioning the retractor and cavity inventory, and dissecting the cystic artery and cystic duct.

The cystic duct will be opened and its catheterization and aspiration of 1 ml of bile will be carried out, which will be sent for analysis of the microbiota.

Subsequently, intraoperative cholangiography will be performed to study the intra and extrahepatic biliary anatomy, followed by anterograde cystic-funicular cholecystectomy and left or right hepatectomy as clinical indication.

## Ethics approval and consent to participate

The study protocol was approved by the Hospital Ethics Committee (Faculty of Medicine, University of São Paulo—FMUSP—CAAE: 29547920.9.0000.0068) and informed consent will be obtained from all individual participants included in the study. The study was registered at ClincalTrials.gov (Identifier: NCT04391426).

None of the transplant donors will be from a vulnerable population and all donors or next of kin will have provided written informed consent that was freely given.

The participant will discuss with Dr. Alberto Meyer or a member of his team about the decision to participate in this study. In this conversation, the purposes of the study will be explained as well as the procedures to be performed, their discomforts and risks, the guarantees of confidentiality and permanent clarifications. It will also be clarified that the participation is free of charge, nor will there be any payments, and that the participant will have guaranteed access to hospital treatment when necessary. The participants will voluntarily agree to participate in this study and will be able to withdraw consent at any time, before or during it, without penalty or loss of any benefit that he may have acquired, or in his service at HCFMUSP. They will sign a consent form and receive a copy initiated by the researcher.

## Processing of samples

Sample collections will be sent to the Molecular Biology Sector of the Central Laboratory Division (LIM-03), which will be responsible for DNA extraction, construction of 16S libraries and sequencing of microbiomes.

**DNA extraction.** DNA extraction will be performed from 1 ml of the content of the e-swab, using the DNA Kit Zymobiomics MiniPrep, according to the manufacturer's recommendations. The DNA will be stored immediately at -20°C until use.

**Sequencing of microbiomes.** The microbiomes will be studied based on two types of sequencing data. The first is the V3/V4 region of the 16S unit of the ribosomal gene; the other is total (shotgun) DNA. In the case of 16S, we will use the 16S Metagenomic Sequencing Library protocol (Illumina, San Diego, USA). In the case of total DNA, Genomic DNA libraries will be built using Nextera® XT DNA Sample Preparation Kit (Illumina) from an input of 1 μg of DNA, according to the manufacturer's recommendations. The 16S and genomic DNA libraries will be sequenced in the MiSeq sequencer (Illumina, San Diego, USA) (or some more modern Illumina sequencer at the time samples become available), using the MiSeq® Reagent Kit v3 (600 cycles; Illumina, San Diego, USA).

**Data analysis.** The data analyses will be performed at the Bioinformatics Laboratory (Setulab), located in the Biochemistry Department, at the Chemistry Institute of the University of São Paulo.

*16S data.* Sequencing the 16S amplicons (bacteria) from the different samples will result in sets of reads. The data will be sent to Setulab servers, where the analyses will be carried out, which will mainly follow the steps and programs available in the QIIME 2 package [14]. Briefly, the steps include quality control, to remove short reads or reads with poor average quality; determination of amplicon sequence variants (ASVs) using the deblur [15] and / or DADA2 [16] programs; taxonomic classification of ASVs; alpha and beta diversity analyses; rarefaction analysis; and associated statistical analyses, seeking to show differences in the microbial composition between control and case samples. We will also determine microbial taxa associated with the main differences using ANCOM [17].

Statistical tests. Wilcoxon tests will be used to compare mean differences between case and control samples for phylum, genus and ASV log abundances. Kruskal-Wallis tests will be performed to compare differences in the means between both groups for alpha diversity. We will use PERMANOVA and ANOSIM [18] to compare beta-diversity differences between groups using three distance metrics: weighted UniFrac, unweighted UniFrac and Bray-Curtis.

*Total DNA data.* these analyses will be carried out according to the following steps:

1. Quality control for the removal of short reads or average quality below a threshold.

2. Quality control to separate DNA from the microbiota from any human DNA that may have been sequenced.

3. Classification of reads using kraken2 [19].

4. Use of the metaWrap pipeline [20] to recover genomes (generation of Metagenome-Assembled Genomes, or MAGs). The application will be separated by sample, and the results will be compared later.

5. Classification of MAGs using the GTDB-tk program [21].

6. Annotation of MAGs by the NCBI PGAP pipeline [22].

7. Evaluation of the representativeness of MAGs in the samples by comparison of taxonomic classification of MAGs with those of reads (step 3).

The results obtained will also be compared with results from the literature, in particular through the eHOMD website [23].

**Data management.** Once this study is completed, the biological material will receive a code and will be filed in the biorepository anonymously, and may be used for other academic studies, without commercial purpose, as long as approved by the Research Ethics Committee, in accordance with the guidelines of the national body that coordinates the principles of research in our country, the National Commission for Ethics in Research.

### Safety and risks regarding adverse events

Bile collection, both in the case group and in the control group, carries minimal risk and discomfort. However, serious adverse events related to examination (ERCP) and / or hepatectomy surgery will be documented on a form throughout the course of the study and will be reported to the principal investigator within 24 hours of observation. If the event is considered to be related to the collection of bile by the principal investigator, he will send a report to the local ethics committee within 3 days.

### Timetable

The research is estimated to last twenty-four months, according to the following schedule in Table 1.

### Statistical analysis

All data will be presented as average and standard deviation. Student's t-test will be performed with SPSS version 20 for Windows and the Mann-Whitney test will be performed using R software and Python scripts. These tests will be used for comparison, as appropriate. A PCA signal will be developed as discriminative analysis.

The sample size calculation, according to the population under investigation, was estimated at 40, with 20 from each group, based on the exposure ratio between cases and controls. In the ratio of 1: 1, with an effect size of 0.98, type I error of 5% and power of 80%. The calculation was based on the distribution of relative abundance between cases and controls, based on the non-parametric Mann-Whitney two-tailed test [24, 25].

For the final calculation, confounding variables (gender, age and body mass index (BMI)) will be considered, as these can influence the variation of the microbiome and, consequently, in its analysis [26].

## Result parameters

The primary purpose of the study is to characterize the specific composition of the biliary microbiota in situations of disease and health. For this, the microbiota of patients with hepatobiliopancreatic diseases will be compared with that of healthy controls. In addition, it seeks to demonstrate the existence of changes in the case of illness.

Further investigation includes disease biomarkers, diagnosis, interventions and therapies in hepatobiliopancreatic diseases.

## Dissemination policy

MICROBILIO results should be presented at international medical conferences on corresponding areas of interest, for example, gastroenterology. Written publications will be

**Table 1. Research schedule.**

| Month | 1 | 2 | 3 | 4 | 5 | 6 | 7 | 8 | 9 | 10 | 11 | 12 | 13 | 14 | 15 | 16 | 17 | 18 | 19 | 20 | 21 | 22 | 23 | 24 |
|---|---|---|---|---|---|---|---|---|---|---|---|---|---|---|---|---|---|---|---|---|---|---|---|---|
| Bibliographic Survey | X | X | X | X | X | X | | | | | | | | | | | | | | | | | | |
| Sample collection | | | | X | X | X | X | X | X | X | X | X | X | X | X | X | X | | | | | | | |
| Sample processing | | | | | | | | X | X | X | X | X | X | X | X | X | X | | | | | | | |
| Data tabulation | | | | | | | | | | | | | | | X | X | X | X | | | | | | |
| Results analysis | | | | | | | | | | | | | | | | | | X | X | X | X | X | | |
| Completion of work | | | | | | | | | | | | | | | | | | | | | | | X | X |

submitted to surgical or endoscopic scientific journals. The authorship of written publications must be confirmed unequivocally by all leading researchers.

## Protocol version

This manuscript refers to the first version of the complete study protocol, made on June 4th, 2020. Modifications to the protocol will be reported to all investigators, the local Research Ethics Committee, all participants and the journal.

## Discussion

The composition and role of the microbiota of the gastrointestinal tract is an increasing focus of study. There is even evidence that these microbes are related to cancer and other chronic diseases [5]. Biliary dysbiosis, for example, has been shown to be a protagonist in the pathogenesis of several diseases in this tract, as has been recently described. However, given the proximity and connection between the pancreatic, hepatic and biliary tracts, it may be that the influence of this microbiota is more extensive, reaching the entire hepatobiliopancreatic tract [27].

On the other hand, the bile microbiota is still little studied, being sequenced in healthy individuals only recently [24]. Little is known about its composition in malignant diseases [28], with biliary lithiasis being the most studied and best characterized disease [24, 29].

This study aims to consolidate existing knowledge and deepen it with additional information. Thus, we seek to broaden the panorama about the different compositions of the biliary microbiota and, possibly, to shed light on a new form of study for hepatobiliopancreatic diseases, as well as strategies on how to deal with such conditions. In the future, more longitudinal studies will be necessary, since those may reveal time-varying compositional changes in the microbiome, both in hepatobiliopancreatic diseases and healthy people.

## Supporting information

**S1 File.**
(DOCX)

**S2 File.**
(DOCX)

## Acknowledgments

The authors are thankful to Justin Axel-Berg for the English corrections and Rossana V. Mendoza López for the statistical analysis.

## Author Contributions

**Conceptualization:** Alberto Meyer.

**Data curation:** Fernanda Sayuri do Nascimento, Milena Oliveira Suzuki, João Victor Taba, Vitoria Carneiro de Mattos, Alberto Meyer.

**Formal analysis:** Fernanda Sayuri do Nascimento, Milena Oliveira Suzuki, João Victor Taba, Vitoria Carneiro de Mattos, Leonardo Zumerkorn Pipek.

**Investigation:** Fernanda Sayuri do Nascimento, Milena Oliveira Suzuki, João Victor Taba, Vitoria Carneiro de Mattos, Leonardo Zumerkorn Pipek, Eugênia Machado Carneiro D'Albuquerque.

**Methodology:** Leonardo Zumerkorn Pipek, Eugênia Machado Carneiro D'Albuquerque, João Carlos Setubal.

**Project administration:** Leandro Iuamoto, Wellington Andraus, João Renato Rebello Pinho.

**Resources:** Leandro Iuamoto.

**Supervision:** Alberto Meyer, Wellington Andraus, João Renato Rebello Pinho, Eduardo Guimarães Hourneaux de Moura, João Carlos Setubal, Luiz Augusto Carneiro-D'Albuquerque.

**Validation:** Alberto Meyer, João Renato Rebello Pinho, Eduardo Guimarães Hourneaux de Moura, João Carlos Setubal, Luiz Augusto Carneiro-D'Albuquerque.

**Visualization:** Eduardo Guimarães Hourneaux de Moura, João Carlos Setubal, Luiz Augusto Carneiro-D'Albuquerque.

**Writing – review & editing:** Alberto Meyer, Luiz Augusto Carneiro-D'Albuquerque.

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
