## [Decision Letter · Decision Letter 0]

6 Oct 2020

PONE-D-20-22918

Analysis of biliary MICRObiota in hepatoBILIOpancreatic diseases compared to healthy people [MICROBILIO]: study protocol

PLOS ONE

Dear Dr. Alberto Meyer,

Thank you for submitting your manuscript to PLOS ONE. After careful consideration, we feel that it has merit but does not fully meet PLOS ONE’s publication criteria as it currently stands. Therefore, we invite you to submit a revised version of the manuscript that addresses the points raised during the review process.  The Reviewers found merit in your study and we encourage resubmisison.

Please submit your revised manuscript within 60 days. If you will need more time than this to complete your revisions, please reply to this message or contact the journal office at plosone@plos.org. Please include the following items when submitting your revised manuscript:

We look forward to receiving your revised manuscript.

Kind regards,

Gianfranco D. Alpini

Academic Editor

PLOS ONE

Journal Requirements:

3.We note that your study will involve tissue/organ transplantation. Please provide the following information regarding tissue/organ donors for transplantation cases analyzed in your study. Even if your study will be retrospective or focused on transplant recipient outcomes, the requested details about the donors are needed to clarify whether the procedures will meet international standards for tissue/organ transplantation.

1. Please state in your response letter and ethics statement whether the transplant cases for this study will involve any vulnerable populations; for example, tissue/organs from prisoners, subjects with reduced mental capacity due to illness or age, or minors.

- If a vulnerable population will be used, please describe the population, justify the decision to use tissue/organ donations from this group, and clearly describe what measures were taken in the informed consent procedure to assure protection of the vulnerable group and avoid coercion.

- If a vulnerable population will not used, please state in your ethics statement, “None of the transplant donors will be from a vulnerable population and all donors or next of kin will have provided written informed consent that was freely given.”

2. In the Methods, please provide detailed information about the procedure by which informed consent will be obtained from organ/tissue donors or their next of kin. In addition, please provide a blank example of the form used to obtain consent from donors, and an English translation if the original is in a different language.

3. Please discuss whether medical costs will be covered or other cash payments will be provided to the family of the donor. If so, please specify the value of this support (in local currency and equivalent to U.S. dollars).

Thank you for your attention to these requests.

Reviewers' comments:

Reviewer's Responses to Questions

**Comments to the Author**

1. Does the manuscript provide a valid rationale for the proposed study, with clearly identified and justified research questions?

Reviewer #1: Partly

Reviewer #2: Yes

2. Is the protocol technically sound and planned in a manner that will lead to a meaningful outcome and allow testing the stated hypotheses?

Reviewer #1: Partly

Reviewer #2: Yes

3. Is the methodology feasible and described in sufficient detail to allow the work to be replicable?

Reviewer #1: Yes

Reviewer #2: Yes

4. Have the authors described where all data underlying the findings will be made available when the study is complete?

Reviewer #1: Yes

Reviewer #2: Yes

5. Is the manuscript presented in an intelligible fashion and written in standard English?

Reviewer #1: Yes

Reviewer #2: Yes

6. Review Comments to the Author

You may also provide optional suggestions and comments to authors that they might find helpful in planning their study.

Reviewer #1: This is a "Registered Report Protocol", where the investigative team plans to conduct an observational, case-control study to understand the composition of the biliary microbiota in patients with hepatobiliopancreatic diseases, compared to health controls. The study design is appropriate, given that well-designed observational studies are often found to be effective, compared to RCTs (randomized controlled trials).

I do have some additional comments that may require attention:

1. Sample size/Power: The sample size/power statement should explicitly mention the "name" of the statistical test used, and the "response" that was used to calculate this. The authors' mentioned effect sizes, but never mentioned the name of the test, whether it was one-tailed, or two-tailed, etc. a two-tailed test. Better to be specific (what test?).

2. Writing style: Even before presenting the sample size/power, I found lot of words and sentences used in describing 16S rRNA method, but I never found "what is the response" they will be using? Microbiome data is usually compositional in nature; see article below:

https://www.frontiersin.org/articles/10.3389/fmicb.2017.02224/full

The writeup needs to clearly describe the type of responses to be used, specifically, are they using the compositional data, or some "summary" of the compositional outcomes to conduct the comparison between disease and controls.

There is a pretty robust literature on conducing microbiome data analysis using compositional data, using standard freeware (like R); the authors are suggested to consider that, versus doing some rudimentary/half-hearted analysis. I strongly suggest addition of a qualified statistician/bioinformatician within the research team to produce nice data analysis results, and not some point and click.

3. Abstract/Statistical Analysis Methods: That brings me to the abstract & statistical analysis proposed. I see names of tests, like Student's t, and Man-Whitney tests. Again, it reads weird, when no responses are mentioned corresponding to the proposed tests, i.e., on which quantities those tests will be used.

4. Longitudinal Design: From the current writeup, study doesn't look longitudinal (correct me if I am wrong!). Plans to consider a longitudinal study maybe crucial (can be part of the Discussion/Conclusion Section), given that those may reveal time-varying compositional changes between the two groups.

Reviewer #2: In the current manuscript, Nascimento et al. reported a registered study protocol aiming at evaluating the difference of microbiota in bile samples between healthy people and those with hepatobiliopancreatic diseases. The authors well described the study procedure in human samples and the processing of DNA preparation for sequencing. However, the authors did not well justify several factors including aging and gender which significantly contribute to the difference of microbiota profiles in their statistical analysis. Based on previous studies (Gut microbiota and aging, Science 04 Dec 2015:Vol. 350, Issue 6265, pp. 1214-1215; Intestinal Microbiota Is Influenced by Gender and Body Mass Index, PLoS One. 2016 May 26;11(5):e0154090.doi: 10.1371/journal.pone.0154090. eCollection 2016.), the proposed statistical analysis which did not consider age, body mass index, and gender as variance factors may not obtain significant outcomes.

7. PLOS authors have the option to publish the peer review history of their article (what does this mean?). If published, this will include your full peer review and any attached files.

Reviewer #1: No

Reviewer #2: No

---

## [Author Response · Author response to Decision Letter 0]

29 Oct 2020

A: Dear reviewers, the necessary style changes were made, according to the requirements of the journal, following the models presented.

A: Dear reviewers, the ethics statement was previously on a separate topic, but it has been modified so that it appears in the Methods section, as requested.

3.We note that your study will involve tissue/organ transplantation. Please provide the following information regarding tissue/organ donors for transplantation cases analyzed in your study. Even if your study will be retrospective or focused on transplant recipient outcomes, the requested details about the donors are needed to clarify whether the procedures will meet international standards for tissue/organ transplantation.

1. Please state in your response letter and ethics statement whether the transplant cases for this study will involve any vulnerable populations; for example, tissue/organs from prisoners, subjects with reduced mental capacity due to illness or age, or minors.

- If a vulnerable population will be used, please describe the population, justify the decision to use tissue/organ donations from this group, and clearly describe what measures were taken in the informed consent procedure to assure protection of the vulnerable group and avoid coercion.

- If a vulnerable population will not be used, please state in your ethics statement, “None of the transplant donors will be from a vulnerable population and all donors or next of kin will have provided written informed consent that was freely given.”

A: Dear reviewers, none of the transplant donors will be from a vulnerable population and all donors or next of kin will have provided written informed consent that was freely given, being duly established in the Ethics statement, as requested.

2. In the ethods, please provide detailed information about the procedure by which informed consent will be obtained from organ/tissue donors or their next of kin. In addition, please provide a blank example of the form used to obtain consent from donors, and an English translation if the original is in a different language.

A: Dear reviewers, in Methods, under the topic Ethics approval and consent to participate, it was informed how consent will be obtained from the organ donors. In this submission, the model of the informed consent form was also attached in its original language (Portuguese) and its English translation in Supporting Infomation.

3. Please discuss whether medical costs will be covered or other cash payments will be provided to the family of the donor. If so, please specify the value of this support (in local currency and equivalent to U.S. dollars).

A: As for costs and payments to donors and families, no payment will be offered and participation will be voluntary. In addition, there will be no medical costs, as all procedures will be performed in the public health system, in the teaching hospital Hospital das Clínicas of the Faculty of Medicine of the University of São Paulo (HCFMUSP), entirely free of charge.

6. Review Comments to the Author

1. Sample size/Power: The sample size/power statement should explicitly mention the "name" of the statistical test used, and the "response" that was used to calculate this. The authors' mentioned effect sizes, but never mentioned the name of the test, whether it was one-tailed, or two-tailed, etc. a two-tailed test. Better to be specific (what test?).

A: Thank you, dear reviewers, for your comment, highlighting this mistake. It has already been corrected and added in Methods, in the sub-item Statistical Analysis, the name of the test, and its due characteristics, as requested, better specifying the details, as the reviewer can check.

2. Writing style: Even before presenting the sample size/power, I found lot of words and sentences used in describing 16S rRNA method, but I never found "what is the response" they will be using? Microbiome data is usually compositional in nature; see article below:

https://www.frontiersin.org/articles/10.3389/fmicb.2017.02224/full

The writeup needs to clearly describe the type of responses to be used, specifically, are they using the compositional data, or some "summary" of the compositional outcomes to conduct the comparison between disease and controls.

There is a pretty robust literature on conducting microbiome data analysis using compositional data, using standard freeware (like R); the authors are suggested to consider that, versus doing some rudimentary/half-hearted analysis. I strongly suggest addition of a qualified statistician/bioinformatician within the research team to produce nice data analysis results, and not some point and click.

A: Thank you for your detailed comments and for providing the reference. As explained in the Methods, our raw data will be sequencing reads of two kinds: rRNA 16S and total (shotgun) DNA. As such, as the reference cited says, these data are compositional. All of the methods and statistical tests that we plan to use take this into account. One of the authors of the present study is Dr. J.C. Setubal, who is the team’s bioinformatician.

3. Abstract/Statistical Analysis Methods: That brings me to the abstract & statistical analysis proposed. I see names of tests, like Student's t, and Man-Whitney tests. Again, it reads weird, when no responses are mentioned corresponding to the proposed tests, i.e., on which quantities those tests will be used.

A: We thank the reviewer for pointing this out. Due to an oversight on our part the Methods section of the abstract was garbled. This has been corrected. For the statistical tests that we plan to use, please refer to the Methods section.

4. Longitudinal Design: From the current writeup, study doesn't look longitudinal (correct me if I am wrong!). Plans to consider a longitudinal study maybe crucial (can be part of the Discussion/Conclusion Section), given that those may reveal time-varying compositional changes between the two groups.

Reviewer #2: In the current manuscript, Nascimento et al. reported a registered study protocol aiming at evaluating the difference of microbiota in bile samples between healthy people and those with hepatobiliopancreatic diseases. The authors well described the study procedure in human samples and the processing of DNA preparation for sequencing. However, the authors did not well justify several factors including aging and gender which significantly contribute to the difference of microbiota profiles in their statistical analysis. Based on previous studies (Gut microbiota and aging, Science 04 Dec 2015:Vol. 350, Issue 6265, pp. 1214-1215; Intestinal Microbiota Is Influenced by Gender and Body Mass Index, PLoS One. 2016 May 26;11(5):e0154090.doi: 10.1371/journal.pone.0154090. eCollection 2016.), the proposed statistical analysis which did not consider age, body mass index, and gender as variance factors may not obtain significant outcomes.

A: We appreciate your suggestion. Longitudinal studies would provide significant information regarding the variation of the microbiome composition of both groups over time, therefore, it is crucial to highlight the need to carry out this kind of study to better understand the microbiota of a site still so little known. This recommendation is best described in the Discussion.

Reviewer # 2's suggestion was also of great value to us to improve and enrich this study protocol. In Methods, in the field of statistical analysis, it is described that these possible confounding variables will be considered, which may, in fact, influence the composition of microbiomes, both in cases and controls.

---

## [Decision Letter · Decision Letter 1]

5 Nov 2020

Analysis of biliary MICRObiota in hepatoBILIOpancreatic diseases compared to healthy people [MICROBILIO]: study protocol

PONE-D-20-22918R1

Dear Dr. Meyer,

We’re pleased to inform you that your manuscript has been judged scientifically suitable for publication and will be formally accepted for publication once it meets all outstanding technical requirements.

Kind regards,

Gianfranco D. Alpini

Academic Editor

PLOS ONE

Additional Editor Comments (optional):

Reviewers' comments:

Reviewer's Responses to Questions

**Comments to the Author**

1. Does the manuscript provide a valid rationale for the proposed study, with clearly identified and justified research questions?

Reviewer #1: Yes

Reviewer #2: Yes

2. Is the protocol technically sound and planned in a manner that will lead to a meaningful outcome and allow testing the stated hypotheses?

Reviewer #1: Yes

Reviewer #2: Yes

3. Is the methodology feasible and described in sufficient detail to allow the work to be replicable?

Reviewer #1: Yes

Reviewer #2: Yes

4. Have the authors described where all data underlying the findings will be made available when the study is complete?

Reviewer #1: Yes

Reviewer #2: Yes

5. Is the manuscript presented in an intelligible fashion and written in standard English?

Reviewer #1: Yes

Reviewer #2: Yes

6. Review Comments to the Author

You may also provide optional suggestions and comments to authors that they might find helpful in planning their study.

Reviewer #1: The authors addressed my previous comments to a greater degree of satisfaction. I have no further comments.

Reviewer #2: Thanks for your responses. All my concerns have been well addressed. I look forward to the outcomes of your study.

7. PLOS authors have the option to publish the peer review history of their article (what does this mean?). If published, this will include your full peer review and any attached files.

Reviewer #1: No

Reviewer #2: No

---

## [Editor Report · Acceptance letter]

10 Nov 2020

PONE-D-20-22918R1 

Analysis of biliary MICRObiota in hepatoBILIOpancreatic diseases compared to healthy people [MICROBILIO]: study protocol 

Dear Dr. Meyer:

I'm pleased to inform you that your manuscript has been deemed suitable for publication in PLOS ONE. Congratulations! Your manuscript is now with our production department. 

Kind regards, 

on behalf of

Dr. Gianfranco D. Alpini 

Academic Editor

PLOS ONE